# Serious Complications and Recurrence following Sacrospinous Ligament Fixation for the Correction of Apical Prolapse

**DOI:** 10.3390/jcm12020468

**Published:** 2023-01-06

**Authors:** Susie De Gracia, Brigitte Fatton, Michel Cosson, Sandrine Campagne-Loiseau, Philippe Ferry, Jean-Philippe Lucot, Philippe Debodinance, Laure Panel, Xavier Deffieux, Olivier Garbin, Géry Lamblin, Caroline Carlier-Guérin, Rajeev Ramanah, Arnaud Fauconnier, Chris Serrand, Xavier Fritel, Renaud de Tayrac

**Affiliations:** 1Service de Gynécologie, Centre Hospitalier Universitaire de Nîmes, Université de Montpellier, 30900 Nîmes, France; 2Service de Gynécologie, CHU Jeanne de Flandre, 59000 Lille, France; 3Service de Gynécologie, CHU d’Estaing, 63000 Clermont-Ferrand, France; 4Service de Gynécologie, CH de La Rochelle, 17000 La Rochelle, France; 5Service de Gynécologie, Hôpital Saint-Vincent-de-Paul, 59000 Lille, France; 6Service de Gynécologie, CH de Dunkerque, 59240 Dunkerque, France; 7Service de Gynécologie, Clinique Beau-Soleil, 34070 Montpellier, France; 8Service de Gynécologie, Hôpital Antoine-Béclère, 92140 Clamart, France; 9Service de Gynécologie, CHU de Strasbourg, 67000 Strasbourg, France; 10Service de Gynécologie, Hôpital Femme-Mère-Enfant, Hôpitaux Civils de Lyon, 69002 Lyon, France; 11Service de Gynécologie, CH de Châtellerault, 86100 Châtellerault, France; 12Service de Gynécologie, Université de Franche-Comté, CHU de Besançon, 25000 Besançon, France; 13Unité de Recherche 7285 “Risques Cliniques et Sécurité en Santé des Femmes et en Santé Périnatale” (RISCQ), UVSQ, Université Paris-Saclay, 78180 Montigny-le-Bretonneux, France; 14Centre Hospitalier Intercommunal de Poissy-Saint-Germain-en-Laye, Service de Gynecologie & Obstétrique, CEDEX, 78303 Poissy, France; 15Laboratoire de Biostatistique, Epidémiologie clinique, Santé Publique Innovation et Méthodologie, Centre Hospitalier Universitaire de Nîmes, 30900 Nîmes, France; 16Service de Gynécologie, Centre Hospitalier Universitaire de Poitiers, 86000 Poitiers, France; 17Université de Poitiers, INSERM CIC 1402, 86021 Poitiers, France

**Keywords:** pelvic organ prolapse, apical prolapse, vaginal approach, sacrospinous ligament fixation, native tissue repair, mesh repair, surgical complication, pelvic organ prolapse recurrence

## Abstract

**Objective:** To evaluate the rates of serious complications and reoperation for recurrence following sacrospinous ligament fixation (SSLF) for apical pelvic organ prolapse. **Methods:** This was a national registry ancillary cohort comparative study. The VIGI-MESH registry includes data from 24 French health centers prospectively collected between May 2017 and September 2021. Time to occurrence of a serious complication or reoperation for genital prolapse recurrence was explored using the Kaplan–Meier curve and Log-rank test. The inverse probability of treatment weighting, based on propensity scores, was used to adjust for between-group differences. **Results:** A total of 1359 women were included and four surgical groups were analyzed: Anterior SSLF with mesh (n = 566), Anterior SSLF with native tissue (n = 331), Posterior SSLF with mesh (n = 57), and Posterior SSLF with native tissue (n = 405). Clavien–Dindo Grade III complications or higher were reported in 34 (2.5%) cases, with no statistically significant differences between the groups. Pelvic organ prolapse recurrence requiring re-operation was reported in 44 (3.2%) women, this was higher following posterior compared with anterior SSLF (*p* = 0.0034). **Conclusions:** According to this large database ancillary study, sacrospinous ligament fixation is an effective and safe surgical treatment for apical prolapse. The different surgical approaches (anterior/posterior and with/without mesh) have comparable safety profiles. However, the anterior approach and the use of mesh were associated with a lower risk of reoperation for recurrence compared with the posterior approach and the use of native tissue, respectively.

## 1. Introduction

Pelvic floor dysfunction is a problem that has a significant impact on a patient’s physical and functional well-being [1]. Pelvic organ prolapse (POP) affects up to 50% of all women, but only 3–6% of them will have symptoms such as vaginal bulge sensation, incontinence, and/or sexual dysfunction [2]. With the increase in life expectancy, it is expected that there will be an increase in the demand for operative interventions to correct these pelvic floor dysfunctions. POP can occur in three vaginal compartments, namely anterior, apical, or posterior, with anterior compartment prolapse being the most common. The apical compartment includes prolapse of the uterus, cervix (following subtotal hysterectomy), or vaginal vault (following total hysterectomy) [3]. It is now well recognized that apical and anterior prolapse are commonly associated; hence, apical suspension is important at the time of anterior wall prolapse correction [4].

Different surgical approaches have been described for apical POP correction, including vaginal, endoscopic, and open procedures. Sacrospinous ligament fixation (SSLF) is considered one of the most common and reference surgical vaginal procedures for the management of apical POP [3,5,6,7]. Initially described by Richter as a vaginal suspension procedure to the sacrospinous ligament, accessed via the para-rectal fossa (posterior approach) [8]. Later, an alternative anterior approach via the para-vesical fossa was described [9]. The use of vaginal meshes for SSLF was introduced in 2002. However, in 2019, the U.S. Food and Drug Administration (FDA) banned the use of vaginal mesh [10]. This resulted in the development of several SSLF modifications using native tissues and utilizing the anterior or vaginal approaches with unilateral or bilateral fixation.

VIGI-MESH is a national registry for the prospective collection of POP surgical treatment outcomes in France, focusing on serious complications and POP recurrence [7,11]. We hypothesized that, compared with the native tissue, the use of mesh in SSLF would be associated with more complications and that anterior SSLF results in less prolapse recurrence compared with the posterior approach. Therefore, the aim of this VIGI-MESH based analysis was to evaluate the rate of serious complications and reoperation for recurrence following apical POP surgical correction using different SSLF techniques.

## 2. Materials and Methods

### 2.1. Study Design

This was a national registry ancillary cohort comparative study. The VIGI-MESH registry includes data from 24 French health centers prospectively collected between May 2017 and September 2021. This registry started in 2017 following its approval by the Comité de Protection des Personnes Ouest III (Institutional Review Board) (IDRBC 2017-A000308-45) and was registered on ClinicalTrials.gov (NCT03052985) in February 2017 [7,11]. Twenty-four French Health Centers participated in this registry.

The ancillary data sought in this registry were all patients who volunteered to participate in the VIGI-MESH registry and who underwent a surgery for the correction of an apical POP by a vaginal approach. The study participants included women who were ≥18 years old presenting with a symptomatic uterine, cervical, or vaginal cuff stage ≥ 2 prolapse according to the pelvic organ prolapse quantification (POP-Q) [12]. The interventions of interest were any of the four different SSLF modifications performed between May 2017 and September 2021, irrespective of the instrument used for suspension (needle holder, Capio^®^ or Capio Slim^®^ (Boston Scientific), i-Stitch^®^ (A.M.I) or Digitex^®^ (Coloplast)). For each of the subjects included in this analysis, the authors verified that the procedure performed was an SSLF performed for an apical POP repair using the specific case report forms [11]. Patients who had a repair that involved combined approaches (anterior and posterior SSLF) and those who had an SSLF in association with another procedure for POP repair, were excluded from this analysis. Data analysis was performed using the VIGI-MESH database on Excel software. Patients were analyzed in four surgical groups: anterior SSLF with native tissue, posterior SSLF with native tissue, anterior SSLF with mesh, and posterior SSLF with mesh.

### 2.2. Outcomes

The main study outcomes were to evaluate the rate of serious complications and failure following SSLF. The Clavien–Dindo classification was used to categorize the surgical complications. Serious complications were defined as a Clavien–Dindo grade III or higher [13], while failure was defined as symptomatic prolapse recurrence requiring reoperation.

### 2.3. Statistical Analysis

The characteristics of the cohort were first described by procedure. Continuous variables were presented as means and standard deviation or medians and interquartile range, while categorical variables as frequencies and associated proportions. When used, appropriate comparison tests were chosen depending on the type of variable and whether the validity conditions were met or not. The Kaplan–Meier curve and log rank test were used for the time to complication and reoperation analyses.

To account for imbalances in cofounding factors between groups, we used the inverse probability of treatment weighting (IPTW), based on the propensity score to adjust for between-group differences [14]. Two propensity scores were calculated with multivariate logistic regression. The first one was for the probability of an anterior versus posterior procedure and the second one was for the probability of a mesh versus native tissue procedure. Potential cofounders in each propensity score were selected according to the initial comparisons between groups (with a *p*-value < 0.20) and clinical consideration. In both scores, the following variables were included: BMI, age, the presence of diabetes, sexual activity, previous hysterectomy, menopause, and the bilaterality of the procedure. For anterior versus posterior score, any other history of surgery was also added, while for mesh versus native tissue, the American Society of Anaesthesiologists (ASA) score was added. The validity of the produced scores and the covariates balance were assessed using standardized mean differences and propensity score histograms. Two outcomes were then analyzed; first, the time until reoperation for recurrence and, second, the time until complication following the initial surgery. For each outcome, IPTW based on the propensity score was used in a Cox proportional hazard model to explore the risk in anterior versus posterior procedures, and then to explore the risk in mesh versus native tissue procedures. The unadjusted hazard ratio and IPTW adjusted hazard ratios were presented with their 95% confidence interval, as well as the corresponding *p*-value. Analyses were conducted with the SAS Enterprise Guide software (version 7.15, SAS Institute Inc., Cary, NC, USA). A two-tailed *p* value < 0.05 was considered statistically significant.

## 3. Results

A total of 1547 women had a vaginal procedure for a symptomatic apical POP in the period between May 2017 and September 2021. One-hundred and eighty-eight participants did not meet the inclusion criteria (63 underwent two types of SSLF at the same time and 125 underwent a McCall Culdoplasty as the apical suspension procedure), resulting in a total of 1359 eligible participants. These included, 566 (41.6%) patients who had anterior SSLF with mesh, 331 (24.4%) anterior SSLF with native tissue, 57 (4.2%) posterior SSLF with mesh, and 405 (29.8%) posterior SSLF with native tissue (Figure 1).

At baseline, the mean age was 69.4 (±9.2) years old, 94% women were menopausal, and the mean body mass index (BMI) was 26.2 (±4.5). The studied participants’ baseline characteristics are presented in Table 1. There were no significant differences between surgical groups, except for menopausal status, mean BMI, and history of previous POP surgery without mesh.

Of the 1359 studied cases, 313 (23%) women had a concomitant procedure, these included hysterectomy in 175 (12.9%) patients and a mid-urethral sling for urinary incontinence in the remaining 138 (10.2%) cases. SSLF was performed bilaterally in 81.1% of the patients and the frequency of this was significantly lower in the group of women who had posterior SSLF with native tissue (*p* < 0.01). The mean surgical time was 76.1 (±31) minutes, with no significant differences between groups, while the mean estimated blood loss was 59.7 (±81.7) mL, with a significantly higher estimated loss after an anterior SSLF with mesh and less blood loss with posterior SSLF with mesh (Table 2). The instruments used for the SSLF were a Capio^®^ or Capio Slim^®^ (Boston Scientific), needle holder, Digitex^®^ (Coloplast), and i-Stitch^®^ (A.M.I), in 52.8%, 17.6%, 15.7%, and 13.5%, respectively. The instrument used was not defined in 0.4% of the study cohort.

The median follow-up time at the time of this analysis was 29.3 months (0.1–55.4 months). Clavien–Dindo grade III or higher complications were reported in 34 cases (2.5%). Five (0.36%) women had a grade IIIa, 28 (2.06%) grade IIIb, and one (0.07%) patient had a grade IVa complication (Table 3). The patient with the grade IVa complication presented with pelvic pain on day 12 postoperative, associated with reno-ureteral dilation and a retro-lateral bladder hematoma requiring intensive care management due to acute renal failure and hemorrhagic shock. The patient was managed conservatively and made a full recovery. A breakdown of complications and their frequencies are presented in Table 4.

Among all of the reported serious complications at 12 months of follow-up, the risk rates were 1.6% (n = 22) and 1.1% (n = 12) in the SSLF with mesh and native tissue, respectively. Figure 2 shows the Kaplan–Meier curves for serious complication with no overlapping and a significant association between groups (*p* = 0.0459). Forty-four women (3.2%) had a reoperation for pelvic organ prolapse recurrences after a SSLF procedure. Figure 3 shows the Kaplan–Meier curve for reoperation for pelvic organ prolapse recurrences after an SSLF that demonstrated no overlapping and a significant difference between the groups with slightly more reoperation for recurrence during follow-up in the posterior SSLF group compared with the anterior SSLF. Based on the propensity score IPTW adjusted models, there was a statistically significant protective effect for the use of mesh compared with native tissue. Similarly, there was a statistically significant protective effect for the anterior SSLF technique compared with the posterior technique regarding the risk of prolapse recurrence (Table 5).

Finally, in order to test the potential effect of different tissue for holding the apical compartment in the right place on the risk of recurrence, we performed a secondary analysis comparing sacrospinous fixation of the cervix (n = 865) versus the vaginal vault (previous or concomitant hysterectomy, n = 618), which did not show a significant difference (log-rank *p*-value = 0.4564).

## 4. Discussion

This ancillary study represents an analysis of the outcomes of a cohort of 1359 women who had one of four SSLF variants for the surgical treatment of their apical POP and who were prospectively included in the VIGI-MESH national registry. Our study demonstrated that this surgical technique is a safe and effective surgical procedure for apical POP treatment. There was a low rate of serious complications. Compared to the posterior approach, anterior SSLF, with mesh or native repair, was associated with a higher frequency of complications (n = 20; 1.47%), but this difference was not statistically significant. All these complications occurred in the postoperative period and none were reported to have occurred intraoperatively. These results concur with those reported in previous studies that used the VIGI-MESH database [11]. There were a total of six vaginal mesh exposures reported in relation to both anterior and posterior SSLF. This constituted <1% of mesh exposure, which is lower than the 8% rate of mesh exposure reported by Maher and associates in their Cochrane review [15]. Some of the complications were related to the concomitant surgery performed, as in the case of four patients who had mid-urethral slings for urinary incontinence and suffered urinary retention or bladder injury. Our results have shown a lower incidence of serious complications after apical POP surgery via the vaginal route than those previously reported in the literature. Indeed, Nager et al. reported 15% of the maximum Dindo score ≥ III [16]. Solomon et al. found that most of the postoperative complications for the anterior bilateral SSLF were voiding difficulties, which were documented in 53 cases (37%) [17]. None of these urinary system injuries were attributable to the SSLF procedure. Yadav et al. compared the most common routes for vaginal apical suspension (SSLF, uterosacral ligament suspension and minimally invasive sacrocolpopexy), and reoperation for complications in the short term after SSLF occurred in 1.2% [18]. Hemorrhage, vaginal bleeding, and hematoma were the most common indications.

The risk of recurrence was higher when the posterior approach was used, irrespective of whether mesh was used or not. This could be explained by the higher risk of postoperative cystocele with the posterior approach. Interestingly, all recurrences occurred in the first year following the procedure. However, this risk seems lower than the previously reported rates [19,20]. The rate of POP recurrence requiring reoperation in our study was 3.2%. In our series, 81.1% of the SSLF were performed bilaterally. Our results are comparable to Salman’s study, which compared unilateral versus bilateral SSLF, reporting recurrence of vaginal vault prolapse in two patients (3.84%) in the unilateral group and no recurrences for the bilateral group, with no statistically significant differences between the two groups with a follow-up of 6 months after surgery [21].

There are several strengths to our work, which increase its internal validity; these include the sample size, the use of prospectively collected data obtained through a national registry, and the use of objectively validated measures to assess our main outcomes. Furthermore, the contribution of several centers to the VIGI-MESH database increase the generalizability of our findings. Nevertheless, we appreciate that the lack of information about surgeons’ experience, the vaginal compartment involved in the recurrence, and long-term follow-up data are limitations to our study. Moreover, being a non-randomized comparison, the presence of demographic and clinical imbalances between our groups is a source of potential bias. However, we tried to mitigate this issue by developing propensity score IPTW adjusted models taking into account relevant possible confounders. Nonetheless, given the low rate of serious complications in this cohort, it is possible that our study was relatively underpowered to detect a significant difference between our analyzed groups.

## 5. Conclusions

According to this large database ancillary study, sacrospinous ligament fixation is an effective and safe surgical treatment for apical prolapse. The different surgical approaches, anterior/posterior and with/without mesh were comparable regarding the rate of serious complications. However, the anterior approach and the use of mesh were associated with a lower risk of reoperation for recurrence compared with the posterior approach and the use of native tissue, respectively.

## Figures and Tables

**Figure 1 jcm-12-00468-f001:**
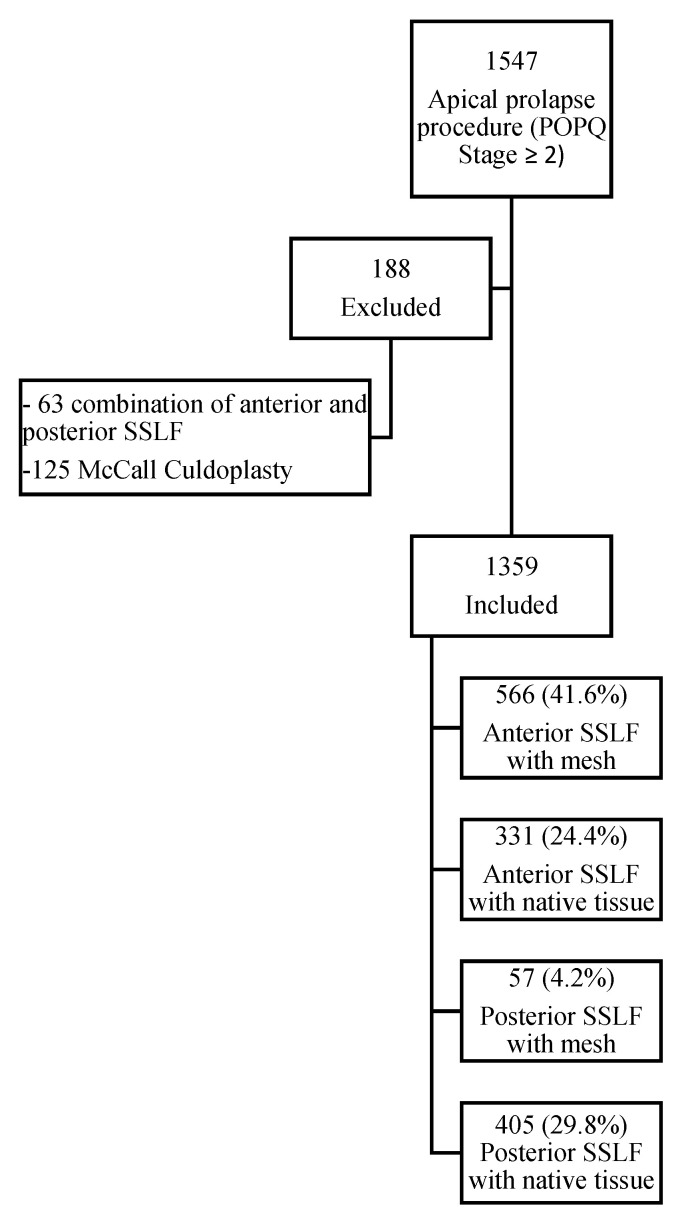
Study flowchart.

**Figure 2 jcm-12-00468-f002:**
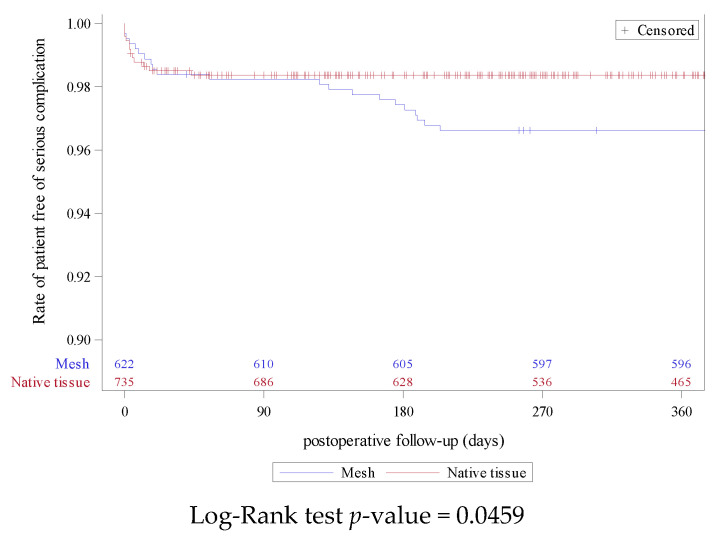
Kaplan–Meier curve for Clavien–Dindo grade IIIa or higher complications comparing SSLF with mesh or with the native tissue.

**Figure 3 jcm-12-00468-f003:**
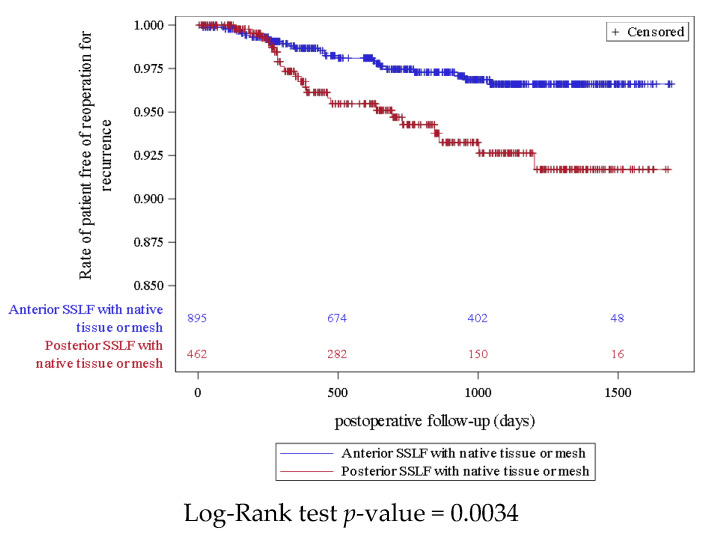
Kaplan–Meier curve for reoperation for pelvic organ prolapse recurrences comparing anterior and posterior SSLF.

**Table 1 jcm-12-00468-t001:** Baseline characteristics for the total population and between the four surgical groups.

	Total Population N = 1359	Anterior SSLF with Mesh N = 566	Anterior SSLF with Native Tissue N = 331	Posterior SSLF with Mesh N = 57	Posterior SSLF with Native Tissue N = 405	*p*
Age mean (SD)	69.4 (9.2)	70.2 (7.3)	69.4 (9.8)	68.1 (7.8)	68.6 (11)	0.22
BMI mean (SD)	26.2 (4.5)	26.3 (4.5)	25.7 (4.3)	27.5 (4.0)	26.4 (4.8)	<0.01
Smoking n (%)	61 (4.6)	21 (3.8)	18 (5.6)	4 (7.0)	18 (4.7)	0.43
Diabetes n (%)	143 (11.1)	73 (13.4)	28 (9.2)	3 (5.4)	39 (10.1)	0.10
Sexually active n (%)	453 (39.0)	174 (35.4)	108 (39.8)	20 (40.0)	151 (43.3)	0.14
Menopausal status n (%)	1261 (94.0)	550 (97.3)	301 (93.8)	56 (98.2)	354 (88.9)	<0.01
ASA score						0.64
1 n (%)	329 (24.2)	135 (23.9)	80 (24.2)	15 (26.3)	99 (24.4)	
2 n (%)	826 (60.8)	346 (61.1)	209 (63.1)	31 (54.4)	240 (59.3)	
3 n (%)	142 (10.4)	68 (12.0)	26 (7.9)	8 (14.0)	40 (9.9)	
4 n (%)	2 (0.1)	2 (0.4)	0	0	0	
Past Surgical History						
Subtotal Hysterectomy n (%)	61 (4.5)	23 (4.1)	9 (2.7)	4 (7.0)	25 (6.1)	0.09
Total Hysterectomy n (%)	273 (20.1)	93 (16.4)	42 (12.7)	17 (29.8)	121 (29.9)	<0.01
SUI surgery with mesh n (%)	110 (8.1)	38 (6.7)	24 (7.3)	7 (12.2)	41 (10.1)	0.13
SUI surgery without mesh n (%)	58 (4.3)	18 (3.2)	9 (2.7)	6 (10.5)	25 (6.2)	<0.01
POP surgery with mesh n (%)	172 (12.7)	62 (11.0)	31 (9.4)	19 (33.3)	60 (14.8)	<0.01
POP surgery without mesh n (%)	132 (9.7)	61 (10.8)	15 (4.5)	7 (12.3)	49 (12.1)	<0.01
Rectal prolapse with mesh n (%)	13 (1.0)	6 (1.1)	2 (0.6)	0	5 (1.2)	0.84
Rectal prolapse without mesh n (%)	22 (1.6)	10 (1.8)	3 (0.9)	1 (1.8)	8 (2.0)	0.63

ASA, American Society of Anesthesiologists patient classification status; BMI, body mass index; POP, pelvic organ prolapse; SD, standard deviation; SUI, stress urinary incontinence.

**Table 2 jcm-12-00468-t002:** Blood loss and surgical time between groups.

	Total Population	Anterior SSLF with Mesh	Anterior SSLF with Native Tissue	Posterior SSLF with Mesh	Posterior SSLF with Native Tissue	*p*
N	1359	566	331	57	405	
Bilateral procedure (%)	1102 (81.1)	564 (99.7)	316 (95.5)	57 (100.0)	165 (40.7)	<0.01
N	1309	549	322	56	382	
Mean surgical Time (±SD)	76.1 (±41)	77.5 (±43.1)	75.3 (±41.6)	64.5 (±36.3)	76.3 (±37.6)	0.12
N	1310	550	321	54	385	
Mean blood loss (±SD)	59.7 (±81.7)	66.4 (±80.5)	53 (±78.1)	21.9 (±32.1)	60.9 (±89.2)	<0.01

**Table 3 jcm-12-00468-t003:** Complications between the four surgical groups according to the Clavien–Dindo classification.

	Total Population N = 1359	Anterior SSLF with Mesh N = 566	Anterior SSLF with Native Tissue N = 331	Posterior SSLF with Mesh N = 57	Posterior SSLF with Native Tissue N = 405
Complications n (%)	34 (2.5)	20 (3.5)	7 (2.1)	2 (3.5)	5 (1.2)
Grade IIIA n (%)	5 (0.4)	5 (0.9)	0	0	0
Grade IIIB n (%)	28 (2.1)	15 (2.7)	6 (1.8)	2 (3.5)	5 (1.2)
Grade IVA n (%)	1 (0.07)	0	1 (0.3)	0	0

**Table 4 jcm-12-00468-t004:** The list of Clavien–Dindo grade IIIa or higher complications and their frequencies (n = 34).

	N	%
Vaginal mesh exposure	6	0.44
Bladder injury	6	0.44
Urinary retention	5	0.37
Ureteral obstruction	4	0.29
Delayed wound of healing or granuloma	3	0.22
Hematoma	2	0.15
Vaginal suture bleeding	2	0.15
Thigh pain	2	0.15
Bladder clot removal	1	0.07

**Table 5 jcm-12-00468-t005:** Multivariate analysis with a propensity score.

	Unadjusted HR [95% CI]	*p*-Value	Adjusted by IPTW HR [95% CI]	*p*-Value
Complication risk				
Mesh vs. Native tissue	2.12 [1.05; 4.26]	0.0359	1.26 [0.73; 2.17]	0.4001
Ant. SSLF vs. Post. SSLF	2 [0.88; 4.59]	0.1004	1.43 [0.88; 2.33]	0.1511
Recurrence risk				
Mesh vs. Native tissue	0.5 [0.27; 0.9]	0.0206	0.42 [0.26; 0.68]	0.0004
Ant. SSLF vs. Post. SSLF	0.43 [0.24; 0.77]	0.0044	0.67 [0.46; 0.99]	0.043

The “unadjusted” models are similar to the analysis via the log rank test, but are done via the univariate Cox model. The “adjusted by IPTW” models are those with adjustment via the calculated propensity score. In both scores, the following variables are included: body mass index, age, the presence of diabetes, sexual activity, previous hysterectomy, menopause, and the bilateral nature of the procedure. For the anterior versus posterior score, any other history of surgery is also added while for the mesh versus native tissue, the ASA score is added.

## Data Availability

Data can be made available upon reasonable request from the corresponding author.

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
