# Peer review of "Serious Complications and Recurrence following Sacrospinous Ligament Fixation for the Correction of Apical Prolapse"

_jcm, 2023, doi:10.3390/jcm12020468_

Round 1

Reviewer 1 Report

The topic of serious complications and recurrence following sacro-spinous ligament fixation closes an interesting gap in the surgical management of vaginal apical POP.

It is to congratulate that the authors gathered national data from different teaching hospitals in France.

The inhomogeneous groups were analyzed with the propensity score IPTW what seems an appropriate method to this research question.

The strengths are the inclusive design, the efficacy and safety parameters. Also, the weaknesses were clearly stated.

I have only two remarks:

- to figure 3 where in the legend the comparators anterior and posterior SSLF should be mentioned (and not mesh and native tissue).

- in the discussion, line 221-224: “Eight per cent of women in the mesh group required repeat surgery for mesh exposure.” Eight per cent was reported in the article of Maher 2016. This French group had only a mesh exposure rate of <1% (6 erosions for 566 anterior und 57 posterior meshes). This should be corrected in the manuscript.

Author Response

We thank the reviewer for his congratulations

Remark 1: to figure 3 where in the legend the comparators anterior and posterior SSLF should be mentioned (and not mesh and native tissue).

We thank the reviewer for this relevant remark: the figure 3 has been replaced

Remark 2: in the discussion, line 221-224: “Eight per cent of women in the mesh group required repeat surgery for mesh exposure.” Eight per cent was reported in the article of Maher 2016. This French group had only a mesh exposure rate of <1% (6 erosions for 566 anterior und 57 posterior meshes). This should be corrected in the manuscript.

We thank the reviewer for this relevant remark: numbers were corrected in the discussion

As suggested by the reviewer, the manuscript has been medically edited, copy edited and proof read again by a native English speaking medical editor.

Reviewer 2 Report

This is very well planned, conducted and written paper with concise and clear description of methods and results.

I don't have any special concerns about the paper but discussion could be a little longer with the description of other studies presenting SSLF results. Especially, those comparing uni- or bilateral fixation of apical compartment. Moreover, data concerning risk of recurrnece should be also divided accordnig to suspension of cervix or vaginal vault, because that might be a difference as different tissuse are "responsible" for holding the apical compartment in the right place.

Author Response

As suggested by the reviewer, discussion was improved, adding 3 new references, including one comparing uni vs bilateral SSFL:

  1. Solomon ER, Marie PS, Jones KA, et al (2020) Anterior Bilateral Sacrospinous Ligament Fixation: A Safe Route for Apical Repair. Female Pelvic Med Reconstr Surg. 26(8): pe33-e36. doi: 10.1097/SPV.0000000000000857
  2. Yadav GS, Gaddam N, Rahn DD (2021) A Comparison of Perioperative Outcomes, Readmission, and Reoperation for Sacrospinous Ligament Fixation, Uterosacral Ligament Suspension, and Minimally Invasive Sacrocolpopexy. Female Pelvic Med Reconstr Surg. 1;27(3):133-139. doi: 10.1097/SPV.0000000000000999
  3. Salman S, Babaoglu B, Kumbasar S et al (2019) Comparison of Unilateral and Bilateral Sacrospinous Ligament Fixa-tion Using Minimally Invasive Anchorage. Geburtshilfe Frauenheilkd. 79(9):976-982. doi: 10.1055/a-0846-5726

Moreover, data concerning risk of recurrence should be also divided according to suspension of cervix or vaginal vault, because that might be a difference as different tissue are "responsible" for holding the apical compartment in the right place.

We thank the reviewer for this relevant remark. As suggested, we have performed a secondary analysis comparing sacrospinous fixation of cervix (n=865) vs vaginal vault (previous or concomitant hysterectomy, n=618) regarding the risk of recurrence, and we did not found a significant difference (log-rank p-value =0.4564). That new result has been add in Results section (without Figure).